# Hyaline Vascular Type of Unicentric Castleman Disease in a Kidney with End-Stage Renal Disease: A Case Report of a Rare Entity at an Unusual Location and a Special Clinical Setting

**DOI:** 10.3390/diagnostics12112878

**Published:** 2022-11-21

**Authors:** Chuan-Han Chen, Hsin-Ni Li

**Affiliations:** 1Department of Radiology, Taichung Veterans General Hospital, Taichung 40705, Taiwan; 2Department of Pathology and Laboratory Medicine, Taichung Veterans General Hospital, Taichung 40705, Taiwan

**Keywords:** castleman disease, ESRD, unicentric castleman disease, multicentric castleman disease, hyaline vascular CD, plasma cell CD

## Abstract

Castleman disease (CD) is an unusual heterogeneous lymphoproliferative disorder that has been classified based on either clinical presentation and disease course or histologic features. Clinically, CD is divided into a unicentric CD (UCD) type and multicentric CD (MCD) type according to the extent of lymph node region involvement and the absence or presence of systemic symptoms. Histologically, it can be categorized into hyaline vascular (HV) type, plasma cell (PC) type and mixed type. The majority of HV-type CD involves a solitary lymph node, and excision surgery is often curative. On the contrary, MCD is a progressive and often fatal disease with lymphadenopathy in multiple nodes, and systemic therapy is needed. Herein we report a unique case of HV-type CD presenting as a single renal mass in a patient with end-stage renal disease (ESRD). Despite the rarity, CD should be included in the differential diagnosis of solitary renal mass lesions. An accurate diagnosis is important to avoid unnecessarily risky or extensive operations.

## 1. Introduction

Castleman disease (CD) (also called angiofollicular lymph node hyperplasia, angiomatous lymphoid hyperplasia and giant lymph node hyperplasia) is an uncommon type of lymphoproliferative disorder with several variant forms. CD is divided into two major types on the basis of the number of enlarged lymph node regions: unicentric and multicentric. Unicentric CD (UCD) involves a single lymph node and often attains a benign clinical course with no or mild symptoms, whereas multicentric CD (MCD) is associated with systemic symptoms and extensive lymph node involvement [1,2]. The great majority of UCD cases can be managed with resection, while MCD cases should be treated with monoclonal antibodies or chemotherapies [3]. Microscopically, CD can be classified into distinct histologic types, namely hyaline vascular CD (HV-CD), plasma cell CD (PC-CD) and mixed-type CD [1,4].

Approximately 80 to 90% of CDs are morphologically categorized as HV type, and mediastinal or abdominal lymph nodes are the most frequently affected sites [2,5,6,7]. Although CD can appear in any part of the body, its origination from renal parenchyma is extremely rare, particularly in an end-stage kidney. It has been reported that patients with end-stage renal disease (ESRD) have a higher risk of developing renal cell carcinoma (RCC) [8,9], so the presence of a solid renal mass might lead to a suspicion of a malignant process preoperatively, and extensive operations in addition to systemic therapies are suggested. In order to avoid unnecessary aggressive management of renal UCD, differentiating from malignant renal neoplasms is important. An accurate diagnosis requires the integration of clinical presentation, imaging and pathological features. Herein, we report a case of HV-UCD confined in the kidney of ESRD. To our knowledge, this is the first case report of HV-UCD deriving from an end-stage kidney.

## 2. Case Description

A 56-year-old female had a medical history of type 2 diabetes for decades with a complication of non-proliferative diabetic retinopathy under treatment with NovoMix 30 FlexPen 24/14 unit, hypertension treated with a combination of antihypertensive agents (e.g., Losartan, Bisoprolol, Hydralazine and Doxazosin) and hyperlipidemia. She suffered from progressive limb edema, shortness of breath, dizziness, nausea and low urine output (50–100 mL × 4 times per day) and was admitted to our hospital under the impression of uremia. Elevated levels of blood urea nitrogen (BUN) (106 mg/dL; normal range: 6–24 mg/dL) and creatinine (17.77 mg/dL; normal range: 0.59–1.04 mg/dL) and a decline of estimated glomerular filtration rate (eGFR) (2 mL/min/1.73 m) were found at our Emergency Department. High random glucose level (195 mg/dL; normal range: 70–140 mg/dL) and increased C-reaction protein (CRP) (5.508 mg/dL; normal range: < 0.3 mg/dL) were noticed, along with anemia and thrombocytopenia (red blood cell count: 3.92 × 10^6^/μL; hemoglobin: 11.0 g/dL; white blood cell count: 5180/μL; and platelet count: 119 ×10^3^/μL). A chest X-ray showed pulmonary edema and left pleural effusion. With the diagnosis of acute renal injury on chronic kidney disease, emergent hemodialysis through a double-lumen catheter was performed. The levels of BUN and creatinine decreased to 44 and 5.29 mg/dL, respectively, after emergent dialysis. A Hickman catheter was subsequently inserted for the purpose of regular dialysis, and the procedure was tolerated well.

During her admission, various imaging studies were conducted. Abdominal ultrasonography (US) (Figure 1) incidentally identified a hypoechoic hypervascular nodule measuring 1.6 cm in the renal parenchyma of the right kidney, by which a neoplasm was suspected. Non-contrast enhanced magnetic resonance imaging (MRI) (Figure 2) was performed in consideration of the dysfunctional status of the kidney. The lesion identified by US was isointense in MRI to the renal parenchyma on both T1- and T2-weighted images, with the appearance of abnormally restrictive diffusion on diffusion-weighted imaging (DWI) and apparent diffusion coefficient (ADC) images. The nature of the lesion was uncertain, but the patient refused further evaluations by then. Eight months later, contrast-enhanced computed tomography (CT) (Figure 3) was undertaken as a survey for future potential transplantation, which displayed no significant increase in lesion size in the presence of early enhancement. Neither abnormal enlargement of regional lymph nodes nor an invasion of the adjacent structures was found, whereas the possibility of renal cell carcinoma could not be completely excluded. She was subsequently admitted for laparoscopic nephroureterectomy.

Grossly, a well-defined unencapsulated tan soft nodular lesion (1.3 × 1.3 × 1.1 cm) was noted in the parenchyma of the lower portion of the kidney. Pathology revealed a well-delineated lesion composed of hyperplastic lymphoid follicles (Figure 4A) with an expanded mantle zone of small lymphocytes and atrophic germinal centers, forming an “onion skin-like” structure (Figure 4B,F). Some of the mantle zones fused and contained more than one germinal center, resulting in a “twinning” feature (Figure 4C). In addition, many of the follicles were penetrated by capillaries and showed a “lollipop” appearance (Figure 4D). In the interfollicular areas, there was an extensive vascular proliferation of high endothelial venules with perivascular hyalinization. Scattered plasma cells dispersed in the interfollicular regions in the absence of sheet aggregation were recognized (Figure 4E). Immunohistochemistry (IHC) stains were performed on selected blocks and showed that the lymphoid follicular centers were immunoreactive for CD20 (Figure 4G), and the thickened mantle zones were simultaneously positive for CD20 and IgD (Figure 4H). In contrast, staining for CD3 (Figure 4I) and Bcl-2 (Figure 4J) in the germinal center and cyclin D1 (Figure 4K) were all negative. CD21 (Figure 4L), which highlighted follicular dendritic cells and revealed proliferations.

Unfortunately, after the surgery, the patient had a stroke involving the territory of the right anterior cerebral artery (ACA) and middle cerebral artery (MCA) with brain edema. Emergent decompressive craniectomy was undertaken. The level of inflammatory marker CRP was 1.165 mg/dL after the surgery. The patient was discharged in a stable condition and exhibited no evidence of neogrowth nor lymphadenopathy in the adjacent tissue after 17 months of follow-up.

## 3. Discussion

CD was first described by Dr. Benjamin Castleman in 1954 when localized and solitary lymphadenopathy occurring in the mediastinum was identified [10]. From then on, several types of CD have been discovered and defined on the basis of clinical, pathological and virological features [1]. According to the clinical presentation and disease course, CD is further divided into UCD, a localized disease involving a single lymph node or anatomic site, and MCD, a systemic and progressive disease with lymphadenopathy in multiple nodes [1,2]. MCD is further subclassified into Kaposi sarcoma herpesvirus- (KSHV, also called human herpesvirus 8 (HHV8)) associated MCD (KSHV-MCD) and KSHV-independent idiopathic MCD (iMCD) [1,11]. Histologically, this rare disease shares characteristic morphologic features, which can be categorized into HV-CD, PC-CD and mixed-type CD. The majority of patients with UCD have HV pathology, while classical MCD is associated with PC or mixed histology, although a proportion of HV morphology has been reported [1,2,12,13,14].

CD has been diagnosed within variable primary sites, including the mediastinal, intra-abdominal, cervical, axillary, retroperitoneal or inguinal lymph nodes [7,15]. Rarely, UCD presents in unusual sites, such as the lungs, orbits, mouth, tonsil, nasopharynx, liver and small intestine [16,17,18,19,20,21,22,23,24,25]. Single-site involvement of renal parenchyma is extremely rare in this disease, with only a little relevant literature published. The cases are listed in Table 1 [5,6,7,26,27,28]. Of the eight cases, there was neither sex predilection (M:F = 4:4) nor a susceptible side of the kidney (Right:left = 5:3). The lesion size ranged from 1.3 to 4 cm. RCC was initially suspected in five cases by imaging studies. All the cases were finally diagnosed as HV-UCD and were free of disease in the subsequent follow-ups after the operations. The incidence of CD with only renal parenchyma involvement might be underestimated given the absence of symptoms, as most cases were detected incidentally during treatment of their underlying diseases or health checkups.

The imaging features of UCD in renal parenchyma have not been well described due to the rarity of this entity, with scant case reports published [5,28,29]. In our case, a hypervascular renal mass was recognized by US. However, the isointensity on both T1WI and T2WI was unusual to the common renal lesions. Rare entities should be included in the differential diagnoses. On contrast-enhanced CT, the mass displayed relatively early and homogeneous enhancement on the arterial phase and equivocal washout enhancement pattern with similar Hounsfield units on the venous phase. The imaging features on contrast-enhanced CT could not exclude the possibility of a malignant neoplasm, such as RCC, particularly in a patient with underlying ESRD. Further extensive clinical work-up and pathological assessment are thus essential to obtain an accurate diagnosis.

Pathologically, the characteristic picture of CD has been well documented [1,3,4]. HV-CD involving lymph nodes displays capsular fibrosis with broad fibrous bands disrupting normal lymph node architecture. An increased number of lymphoid follicles is present. Mantle zones are hyperplastic and composed of concentric rings of small lymphoid cells surrounding atrophic germinal centers depleted of B cells (so-called “onion skin pattern”). Hyaline deposits and follicular dendritic cells remain in these atrophic germinal centers. The depleted germinal centers are penetrated by hyalinized blood vessels (so-called “lollipop lesions”). The interfollicular zone is infiltrated by scattered plasma cells, immunoblasts and eosinophils within the stroma in the absence of sheets of plasma cells, which are normally present in PC-CD [1,4]. It is noteworthy that before establishing a diagnosis of CD, several lesions with prominent HV-CD-like changes, such as reactive follicular hyperplasia and lymphoid neoplasms, mantle cell lymphoma, follicular lymphoma and nodal marginal zone lymphoma, should be excluded [1,4,5]. Immunohistochemistry staining acts as an easy and useful tool to differentiate the aforementioned entities [5]. Our case revealed typical histomorphological features of HV-CD, except for the absence of a fibrous capsule between the lesion and renal parenchyma. The disappearance of the capsule and fibrotic bands might result from different anatomic sites and distinct tissue components.

**Table 1 diagnostics-12-02878-t001:** Clinical, radiological and pathological features of UCD occurring in renal parenchyma.

Case No.	Age/Sex	Symptom	Comorbidities	Side	Size	Imaging	Type	Treatment	Follow-Up	Reference
1	56/M	None Detected in a health checkup	HTN and diabetes	Right upper-middle	4 × 3.5 cm	CT: Iso-density lesion	HV	Partial nephrectomy	Free of disease 6 months after surgery	[5]
2	61/M	None Detected in a health checkup	None	Right upper	2 × 1.8 cm	MRI: Lesion detected CT: Enhancement at the early phase and washed out at the late phase, suspecting RCC	HV	Laparoscopic partial nephrectomy	NA	[28]
3	59/F	None Incidentally found	Pneumonia	Left posterior midportion	1.5 cm	CT and MRI: Enhancing mass, suspecting RCC	HV	Open partial nephrectomy	NA	[27]
4	NA/F	Low-grade fever, sweating, malaise and polyarthralgia	NA	Right	NA	Renal mass	HV	Nephrectomy	Free of disease 12 months after surgery	[26]
5	69/F	NA	None	Right inferior pole	2.0 cm	CT: Renal mass, suspecting RCC, leiomyoma or angiomyolipoma	HV	NA	NA	[7]
6	38/M	None Incidentally found	Renal stones and degenerative disk disease	Left anterior mid-region	1.8 cm	CT: Enhancing renal mass	HV	Open partial nephrectomy	Free of disease 10 months after surgery	[6]
7	70/M	None Incidentally found	Chronic diverticulitis	Left	2.0 cm	CT: Enhancement at the early phase and washed out at the late phase, suspecting RCC and other malignant tumors MRI: Iso-intensity in the T1-weighted image and low-intensity in the T2-weighted image	HV	Partial nephrectomy	Free of disease 8 months after surgery	[29]
8	56/F	None Incidentally found	ESRD, HTN, diabetes, hyperlipidemia	Right lateroposterior cortex of the lower pole	1.3 cm	US: Lesion detected MRI: Iso-intensity in the T1- and T2-weighted images and low ADC value CT: Early enhancement, suspecting RCC	HV	Laparoscopic nephroureterectomy	Free of disease 17 months after surgery	Our case

Abbreviation: ADC, apparent diffusion coefficient; CT, computed tomography; ESRD, end-stage renal disease; F, female; HTN, hypertension; HV, hyaline vascular; M, male; MRI, magnetic resonance imaging; NA, not available; No., number; RCC, renal cell carcinoma; US, ultrasonography. Unlike MCD, which has been believed to be associated with viral infections and an elevated level of a multifunctional cytokine, IL-6, the etiology of UCD has not yet been completely clarified [1,11]. It has been suggested that a neoplastic process and dysplastic change involving the follicular dendritic cells play a central role [1,30,31,32]. The presence of these accessory cells of the immune cells in renal parenchyma is unexpected but may reflect an underlying inflammatory process with the recruitment of lymphoid follicles. The occurrence of UCD in our patient with ESRD, a disease with an evident long-standing chronic inflammatory status [33], might support the phenomenon of dysregulated immune responses. This assumption may be further confirmed by the other cases listed in Table 1 with other comorbidities, such as renal stones (case no. 6) and diabetes (case no. 1). Additional case studies are needed to help better define the etiology of this disease.

Treatment guidelines for UCD have been standardized by the international consensus and published recently [3], in which complete surgical excision is the optimal therapy where unresectable asymptomatic UCD may be observed. The great majority of UCD cases can be managed with resection, resulting in overall survival of > 90% at 5 years [3]. However, the guidelines indicate the choices of treatment recommended on the basis of UCD involving lymph nodes rather than unusual locations. It is unknown whether resection is still essential for UCD located in other anatomical sites, which needs further studies.

## 4. Conclusions

UCD that solely involves the kidney is an exceptionally rare occurrence but should be included in the differential diagnosis of solitary renal mass lesions in patients with ESRD. Becoming familiar with its clinical, radiological and pathological characteristics prompts an accurate diagnosis to avoid unnecessary systemic and aggressive management.

## Figures and Tables

**Figure 1 diagnostics-12-02878-f001:**
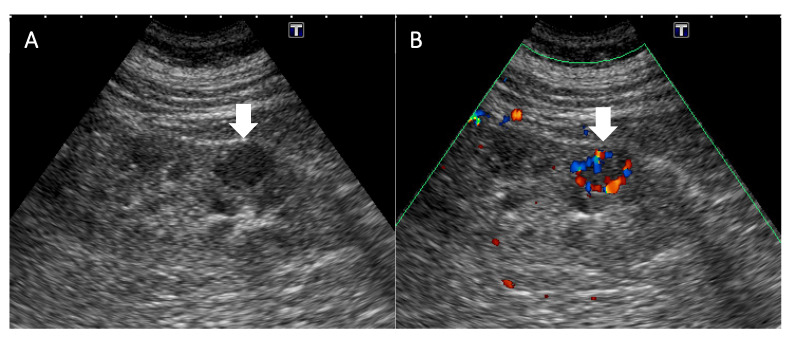
Ultrasonography of this case. (**A**) Ultrasonography reveals a hypoechoic nodule in the renal parenchyma of the right kidney (arrow). (**B**) Color Doppler imaging shows increased peripheral vascularity of the lesion (arrow).

**Figure 2 diagnostics-12-02878-f002:**
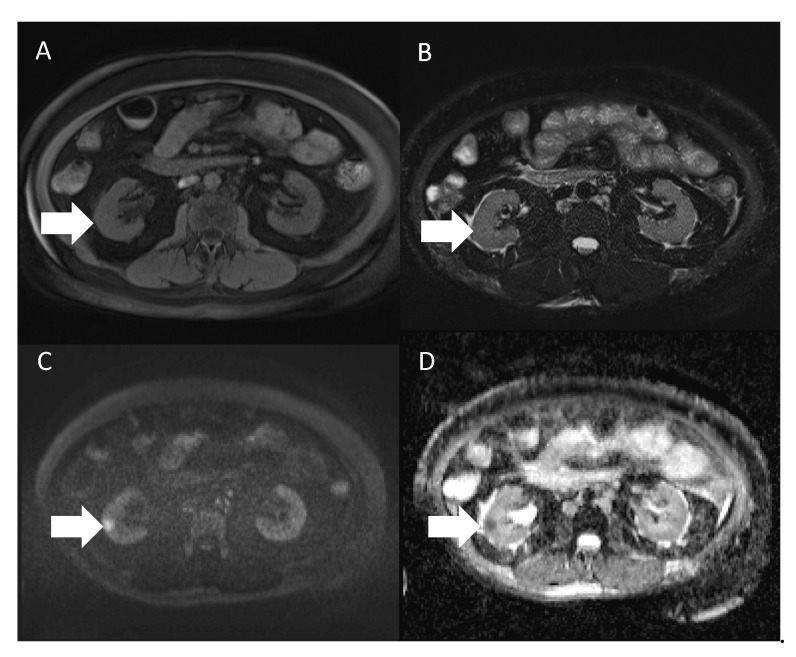
Non-contrast enhanced magnetic resonance imaging (MRI) of this case. The lesion (arrow) shows relatively isointense on both T1-weighted imaging with fat saturation (**A**) and T2-weighted imaging (**B**), with high signal intensity on diffusion-weighted imaging (**C**) and low-value appearance on apparent diffusion coefficient imaging (**D**).

**Figure 3 diagnostics-12-02878-f003:**
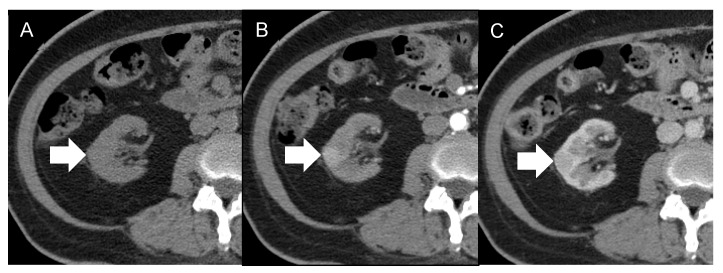
Axial computed tomography (CT) with dynamic contrast enhancement of the case. The lesion (arrow) is relatively isodense on pre-contrast CT imaging (**A**), relatively early enhancement on arterial phase (**B**), and equivocal washout enhancement pattern with similar Hounsfield units on venous phase (**C**) as those on arterial phase. Neither abnormal enlargement of regional lymph nodes nor invasion of the adjacent structures is found.

**Figure 4 diagnostics-12-02878-f004:**
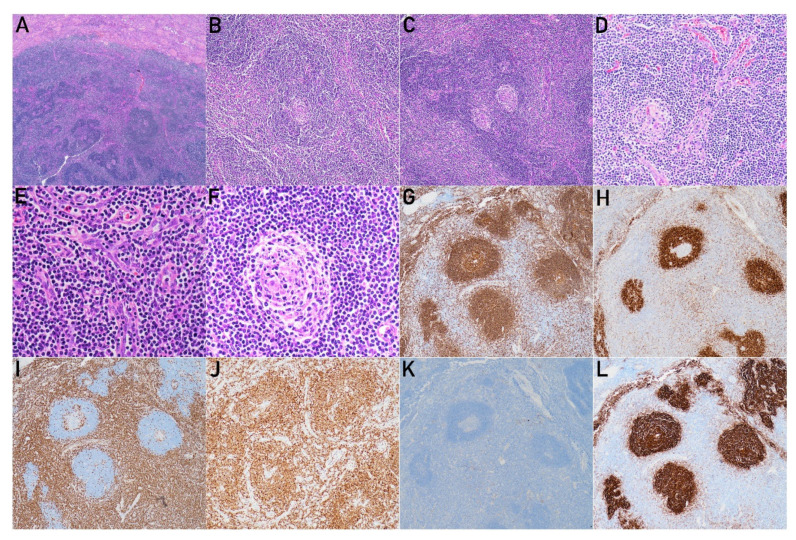
Pathological features of this case. (**A**) A well-delineated lesion composed of hyperplastic lymphoid follicles distinct from underlying renal parenchyma (H&E stain, ×20). (**B**,**F**) Thickened mantle zone and atrophic germinal centers, forming an “onion skin-like” structure (H&E stain, ×100 and ×400, respectively). (**C**) Twinning feature (H&E stain, ×100). (**D**) The follicle is penetrated by hyalinized capillaries (H&E stain, ×200). (**E**) The extensive vascular proliferation of high endothelial venules with perivascular hyalinization in the interfollicular regions (H&E stain, ×400). (**G**) Immunohistochemistry of CD20 highlights the mantle zones and germinal centers (×100). (**H**) Immunohistochemistry of IgD highlights the mantle zone (×100). (**I**) Immunohistochemistry of CD3 highlights the T lymphocytes (×100). (**J**) Immunohistochemistry of Bcl-2 is expressed in T cells and mantle cells but not germinal centers (×100). (**K**) Immunohistochemistry of cyclin D1 shows negative staining (×100). (**L**) Immunohistochemistry of CD21 highlights the proliferation of follicular dendritic cells (×100).

## Data Availability

Not applicable.

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
