# Peer review of "Hyaline Vascular Type of Unicentric Castleman Disease in a Kidney with End-Stage Renal Disease: A Case Report of a Rare Entity at an Unusual Location and a Special Clinical Setting"

_diagnostics, 2022, doi:10.3390/diagnostics12112878_

Round 1

Reviewer 1 Report

The authors present an interesting case of unicentric castleman disease of the kidney in a patient with ESRD. The case report is well-written, however, some changes should be implemented in order to better highlight the importance of the case. Since Castleman disease of the kidney without lymph node involvement is a rare phenomenon, authors should aim to include as much information about their case and experience as possible.

In the section “Case description”, authors should give more information regarding the patient’s overall health status and history; glucose levels, hemoglobin levels, WBC count upon admission etc. are vital in order to give readers a better understanding of the patient’s condition at the time. They should also mention the stage of the patient’s chronic kidney disease (CKD), as well as associated parameters, such as GFR, serum urea and creatinine levels.

Furthermore, medications the patient was receiving for regulating her diabetes, hyperlipidemia, hypertension etc. should be mentioned, as well as their potential role in the acute renal injury (AKI) the patient suffered, on the grounds of CKD. Other forms of treatment, such as dialysis, should also be mentioned, if applicable. What was the therapeutic approach regarding the patient's AKI?

It is stated that the patient underwent a CT scan, as part of a "pre-transplant survey"; the patient was later admitted for laparoscopic nephroureterectomy. Please elaborate regarding "pre-transplant survey"; was the patient already a potential kidney transplant candidate on her initial admission? Was a transplant performed directly after the nephroureterectomy? If so, please provide information regarding the type of the transplant and the surgical approach. Why wasn't a CT guided biopsy of the lesion performed prior to the surgery (as part of the pre-transplant process), upon suspicion of an RCC, in order to determine the patient's suitability for a transplant?

Regarding the patient’s postoperative recovery and follow-up period, authors should highlight the change of the aforementioned biochemical parameters and present the current status of the patient (renal function tests, need for dialysis etc.) Authors should clearly describe what would constitute a recurrence of the disease, after a nephrouretererectomy; novel mass, enlarged lymph nodes in adjacent parenchyma?

The patient in the current case report denied further investigation of the renal mass; however, authors highlight the importance of an accurate diagnosis, in order to avoid unnecessary surgical procedures. Given their initial suspicion of an RCC, and with CT’s inconclusive evidence, it would be useful to present the workup that guidelines suggest (specific cancer markers, biopsy etc.) in order to limit the differential diagnosis.

Reviewer 2 Report

Very interesting case report. Can you tell us more more about inflamatory markers (CRP, fibrinogen...) as it may help to differentiate UCD and MCD.
